# Fecal Volatile Metabolomics Predict Gram-Negative Late-Onset Sepsis in Preterm Infants: A Nationwide Case-Control Study

**DOI:** 10.3390/microorganisms11030572

**Published:** 2023-02-24

**Authors:** Nina M. Frerichs, Sofia el Manouni el Hassani, Nancy Deianova, Mirjam M. van Weissenbruch, Anton H. van Kaam, Daniel C. Vijlbrief, Johannes B. van Goudoever, Christian V. Hulzebos, Boris. W. Kramer, Esther J. d’Haens, Veerle Cossey, Willem P. de Boode, Wouter J. de Jonge, Alfian N. Wicaksono, James A. Covington, Marc A. Benninga, Nanne K. H. de Boer, Hendrik J. Niemarkt, Tim G. J. de Meij

**Affiliations:** 1Department of Pediatric Gastroenterology, Emma Children’s Hospital, Amsterdam Gastroenterology Endocrinology Metabolism Research Institute, Amsterdam UMC, 1105 AZ Amsterdam, The Netherlands; 2Department of Neonatology, Amsterdam Reproduction and Development Research Institute, Emma Children’s Hospital, 1105 AZ Amsterdam, The Netherlands; 3Department of Neonatology, University Medical Center Utrecht, Wilhelmina Children’s Hospital, 3584 CX Utrecht, The Netherlands; 4Department of Neonatology, Beatrix Children’s Hospital, University Medical Center Groningen, 9713 GZ Groningen, The Netherlands; 5Department of Pediatrics, Maastricht University Medical Centre, 6229 ER Maastricht, The Netherlands; 6Department of Neonatology, Isala Hospital, 8025 AB Zwolle, The Netherlands; 7Department of Neonatology, University Hospitals Leuven, 3000 Leuven, Belgium; 8Department of Neonatology, Radboud UMC, Amalia Children’s Hospital, 6525 XZ Nijmegen, The Netherlands; 9Tytgat Institute for Liver and Intestinal Research, Amsterdam Gastroenterology Endocrinology Metabolism Research Institute, Amsterdam UMC, University of Amsterdam, 1105 AZ Amsterdam, The Netherlands; 10School of Engineering, University of Warwick, Coventry CV4 7AL, UK; 11Department of Gastroenterology and Hepatology, Amsterdam Gastroenterology Endocrinology Metabolism, Amsterdam University Medical Centre, Vrije Universiteit Amsterdam, 1081 HZ Amsterdam, The Netherlands; 12Department of Neonatology, Máxima Medical Center, 5504 DB Veldhoven, The Netherlands

**Keywords:** neonatology, volatile organic compounds, gas chromatography—ion mobility spectrometry, gas chromatography—time of flight—mass spectrometry, fecal biomarker, microbiota

## Abstract

Early detection of late-onset sepsis (LOS) in preterm infants is crucial since timely treatment initiation is a key prognostic factor. We hypothesized that fecal volatile organic compounds (VOCs), reflecting microbiota composition and function, could serve as a non-invasive biomarker for preclinical pathogen-specific LOS detection. Fecal samples and clinical data of all preterm infants (≤30 weeks’ gestation) admitted at nine neonatal intensive care units in the Netherlands and Belgium were collected daily. Samples from one to three days before LOS onset were analyzed by gas chromatography—ion mobility spectrometry (GC-IMS), a technique based on pattern recognition, and gas chromatography—time of flight—mass spectrometry (GC-TOF-MS), to identify unique metabolites. Fecal VOC profiles and metabolites from infants with LOS were compared with matched controls. Samples from 121 LOS infants and 121 matched controls were analyzed using GC-IMS, and from 34 LOS infants and 34 matched controls using GC-TOF-MS. Differences in fecal VOCs were most profound one and two days preceding *Escherichia coli* LOS (Area Under Curve; *p*-value: 0.73; *p* = 0.02, 0.83; *p* < 0.002, respectively) and two and three days before gram-negative LOS (0.81; *p* < 0.001, 0.85; *p* < 0.001, respectively). GC-TOF-MS identified pathogen-specific discriminative metabolites for LOS. This study underlines the potential for VOCs as a non-invasive preclinical diagnostic LOS biomarker.

## 1. Introduction

Preterm infants are at increased risk of developing late-onset sepsis (LOS) because of their immature immune system. LOS results in significant mortality rates and is associated with long-term neurocognitive deficits [1,2,3,4,5,6]. Incidence rates are reversely correlated with gestational age and birthweight, and range from 12% to 40% [1,2,3,7,8,9,10,11,12]. Recognition of LOS may be challenging since early-stage symptoms are commonly subtle and non-specific. To date, the gold standard for diagnosis of LOS is a positive blood culture. However, the sensitivity of this test is impaired because often, only small volumes of blood can be obtained from preterm neonates [13,14]. In addition, blood cultures are invasive, prone to contamination and have a 48–72 h laboratory turnaround time [3,4,15,16]. Since early detection and adequate initiation of targeted antibiotic treatment are important prognostic factors for LOS, there is a yet unmet need for early diagnostic and predictive biomarkers [17].

Volatile organic compound (VOC) analysis is a non-invasive technique that can monitor alterations of the host’s cellular metabolism and the gut microbiota [18,19,20]. Different VOC measuring techniques on various sample types have been employed to detect and predict diseases over the last decade, including variants of mass spectrometry (MS), such as liquid chromatography-MS (LC-MS) and the portable electronic nose (e-nose) [21]. The generated data is often analyzed by advanced machine learning techniques, resulting in accurate detection of inflammatory bowel disease, Alzheimer’s disease, preterm birth, and several types of cancer [22,23,24,25,26,27,28,29,30]. Recently, VOC-evoked neuronal fingerprints generated by insect brains were used as a biological brain-based pattern-sensing technique for detection of cancer in vitro [31]. The rapid advancements in VOC detection methods and machine learning open up a wide research area with great opportunities for non-invasive, early disease detection.

The potential of fecal VOCs as early, non-invasive biomarkers for LOS has been recognized in different studies [32,33,34]. Fecal VOCs are largely produced by the gut microbiota and its interaction with the host; they are therefore considered to reflect microbial composition and functional activity [35]. Our research group previously measured VOC profiles using two pattern recognition techniques––a 32-sensor e-nose and high field asymmetric waveform ion mobility spectrometry (FAIMS). Profound differences were described between LOS and controls up to three days before clinical onset of the disease [32,33,34].

In the current study, we aimed to test the potential of two complementary sensitive techniques for LOS prediction in a novel validation cohort. Gas chromatography—ion mobility spectrometry (GC-IMS) was used for fecal VOC pattern recognition, and gas chromatography—time of flight—MS (GC-TOF-MS) for identification of unique fecal volatile metabolites.

## 2. Materials and Methods

### 2.1. Subjects

This study was embedded in a large ongoing prospective cohort study in the Netherlands and Belgium. Inclusion criteria were birth <30 weeks of gestation and hospitalization in one of nine participating neonatal intensive care units during (part of) the first 28 days of life. Infants with congenital gastrointestinal anomalies were excluded. For the current multicenter case-control study, additional exclusion criteria were early-onset sepsis, necrotizing enterocolitis (NEC) Bell’s stage ≥ 2A and spontaneous intestinal perforation. Additionally, infants that had participated in other sub-studies on VOCs were excluded.

Infants diagnosed with LOS within the first 28 days of life were included when all Vermont Oxford criteria for LOS were met: (1) clinical signs of generalized infection, (2) a positive blood culture ≥72 h after birth, and (3) initiation of antibiotic treatment with the intention to treat for ≥5 consecutive days [36]. Only the first episode of LOS was considered. Clinical onset of LOS was defined as the postnatal age at which the positive blood culture sample was drawn. Infants with coagulase-negative staphylococci (CoNS) isolated from the blood culture were only included if the CRP level was ≥10 mg/L to limit the risk of including infants with contaminated blood cultures. Controls were eligible to participate if they did not fulfil the Vermont Oxford criteria for LOS during the study period or if case CRP levels did not exceed 10 mg/L during the study period. Every LOS case was matched to a control infant, based on the center of birth, gestational age (±2 days), birthweight (±150 g), and postnatal age at LOS onset (±0 days). All LOS cases and controls needed at least one fecal sample in the three days prior to clinical LOS onset to be included.

### 2.2. Data and Sample Collection

Fecal samples and clinical data were collected daily from all included subjects, from birth up to 28 days of life. Feces was scooped from diapers by the nursing staff, placed in a container (Stuhlgefäß 10 mL, Frickenhausen, Germany), and subsequently stored at −20 °C. Demographic and baseline clinical data were collected from electronic patient records (EPD). Feeding type was based on the following proportions of human and formula milk (HM and FM, respectively): (1) Predominant HM feeding was defined as >80% of average daily enteral intake consisting of HM, (2) Predominant FM feeding as >80% of average daily enteral intake consisting of FM, and (3) combined HM and FM with average enteral intake of both HM and FM between 20–80%. No distinction was made between mother’s own milk and donor breastmilk.

Before shipment to the laboratory, collected samples were transferred (frozen) into sterile glass vials (20 mL headspace vial, Thames Restek, Saunderton, UK), and shipped on dry ice (−78.5 °C) to the School of Engineering at the University of Warwick (Coventry, UK) for VOC analysis, where they were stored at −20 °C until further handling.

### 2.3. Fecal VOC Analysis

Samples from cases with LOS were analyzed both together and separately based on the observed LOS-causing pathogens:Gram-positive LOS (excluding CoNS): Subcategory *Staphylococcus aureu*s (*S. aureus*) LOS.CoNS-LOSGram-negative LOS: Subcategory *Escherichia coli* (*E. coli*) LOS

For all categories together and for each separate subgroup, analyses were performed for all time points combined (t_−1–(−3)_), and at three separate time points: (1) one (t_−1_), (2) two (t_−2_), and (3) three (t_−3_) days prior to LOS onset. The corresponding postnatal age was analyzed in controls. In every subgroup, the LOS cases were compared with their matched controls.

Two different techniques were used for fecal VOC analysis. First, all samples were analyzed using GC-IMS, which is based on pattern recognition. A subgroup of fecal samples was additionally analyzed using GC-TOF-MS, which is a lab-based un-targeted metabolomics approach for the detection and identification of unique metabolites in the headspace (emitted VOCs from samples) of fecal samples by measuring its mass-to-charge ratio. The samples for GC-TOF-MS were selected based on availability of sufficient samples and sample mass following GC-IMS analysis. For GC-TOF-MS, only infants who provided at least two fecal samples within three days prior to LOS onset, or at the corresponding postnatal days in controls, with a sample weight of ≥150 mg per sample (cases and controls), were included. This was done to ensure sufficient headspace concentration. Fecal VOC analysis was performed in random order.

#### 2.3.1. Fecal VOC Analysis Using GC-IMS

The GC-IMS device (GC-IMS, FlavourSpec^®^, G.A.S., Dortmund, Germany) was fitted with a CTC PAL autosampler (CTC, Zwingen, Switzerland) [37]. The samples were analyzed according to the protocol described by Rouvroye et al. [38]. Fecal samples were heated inside the vials for eight minutes at 80 °C to generate sufficient headspace concentration. The molecules in the headspace were then injected into the GC-IMS instrument and first pre-separated by retention time based on chemical interactions with the GC column. After entering the ion mobility spectrometer, the molecules were ionized by a low-radiation tritium (H3) source, creating reactant ions. Subsequently, these ions were moved to an electric field at atmospheric pressure against the flow of an inert drift gas, creating nitrogen/ion collisions. These collisions result in the ions selectively losing momentum, extending the travel time through the drift-tube before being detected. Thus, the ions are separated based on a combination of their mass, charge, and size. The ion current is detected by an electrometer as a function of time. For the experiments, the selected conditions were as follows: the GC was a 15 m, SE-54 column (CS Chromatographie, Langerwehe, Germany) and the analysis was performed with the GC heated to 45 °C using nitrogen 99.9% (3.5 bar) as the carrier gas. The IMS also used nitrogen as the drift gas, and was performed at 45 °C. The flow rate for the nitrogen into the GC was 20 mL/min (34.175 kPa) for six minutes, and the drift tube flow rate was 150 mL/min (0.364 kPa) (IMS).

#### 2.3.2. Fecal VOC Analysis using GC-TOF-MS

The GC-TOF-MS analysis was conducted using an Rxi-624Sil MS column (length 20 m, internal diameter 0.18 mm, thickness 1.0 μm; Thames Restek, Saunderton, UK) coupled to a Markes Bench TOF-HD (Markes International, Bridgend, UK) and a Markes International TD-100 and Unity-xr thermal desorption unit. The headspace was first loaded from the vials onto bio-monitoring sorbent tubes from Markes (C2-AAXX-5149). This was done by heating the samples for 20 min at 40 °C before pumping the headspace into the tubes for 20 min (5 ml/minute) using an SKC Pocket Pump (SKC Ltd., Dorset, UK). The sorbent tubes were loaded into the auto-sampler. Markes TOF-DS software (Version 4.5.1) was used to add the unique identifier codes to the tubes and select the appropriate run sequence.

Subsequently, the headspace was pre-separated using GC before being injected into the TOF-MS transfer line. The molecules were ionized and accelerated using an electric field, resulting in the same kinetic energy for ions with a similar charge. The velocity of the ions is also determined by their mass (heavier ions with the same charge move slower). The time to reach the detector at a set distance was measured.

The GC settings were as follows: the standby split of 150 °C ran with an overlap (to reduce overall run time), and the GC temperature increased from 40 °C to 280 °C by increments of 20 °C per minute. The desorption was performed by pre-purging the sample for one minute and then heating it for 10 min at 250 °C with a trap purge time of one minute. Subsequently, the trap was cooled to 30 °C and then purged for three minutes at 300 °C. Other settings were as follows: the filament voltage was set to 1.7 V (10-s filament delay), the transfer line was set to 250 °C, the ion source temperature was 250 °C, and electron ionization was performed at −70 V. Masses from 35 to 350 atomic mass units were analyzed.

### 2.4. Statistical Methods

#### 2.4.1. Sample Size Calculation

Sample size calculation for this validation study was based on the results of a previous study on fecal VOC analysis for early detection of LOS [33]. In that study, using the high-field asymmetric waveform ion mobility spectrometry (FAIMS) method, 121 LOS cases and 121 matched controls were included. VOC patterns from controls and cases were compared up to 3 days preceding LOS. The area under the curve (AUC) obtained to discriminate LOS from controls, caused by gram-positive and gram-negative pathogens, ranged from 0.69 to 0.87, respectively. We aimed to obtain a diagnostic accuracy of approximately 0.85 for both gram-positive and gram-negative LOS for the current study. To obtain an AUC of 0.85 with a confidence interval (CI) of 0.1, inclusion of at least 117 LOS cases and 117 matched controls was required.

#### 2.4.2. Demographic Data

For statistical analyses of demographic and clinical data, IBM SPSS^®^ version 24 (Armonk, NY, USA) was used. A χ^2^ test, an independent *t*-test, or a non-parametric test were used to calculate the *p-*values, as appropriate, with *p-*values < 0.05 considered significant.

#### 2.4.3. Pre-Processing GC-IMS Data

Two pre-processing steps were included for the GC-IMS analysis. This was to reduce the dimensionality of the dataset and make subsequent data analysis less computationally heavy. The GC-IMS method produces datasets with a high number of data points per sample (11 million), but with much lower information content. To reduce the dimensionality, the central section of the data was cropped to only retain relevant sample information. Then, a threshold was applied to remove background noise, reducing the dimensionality to approximately 10,000 data points. The same crop parameters and threshold were used on all the samples and were selected manually by analyzing the raw data.

#### 2.4.4. Pre-Processing GC-TOF-MS Data

During the analysis of the samples, the TOF-DS™ software (Version 4.5.1) applied dynamic background compensation, which automatically removes chromatogram background interference. This software also integrates and deconvolutes the peaks in the chromatogram. Subsequently, the compounds that were present were identified. The integration settings were as follows: Global Height Reject: 10,000; Global Width Reject: 0.001; Baseline Threshold: 3; and Global Area Reject: 10,000. The NIST (National Institute of Standards and Technology) database was used to identify specific compounds. Both forward and reverse were matched with a minimum match factor of 450.

#### 2.4.5. Class Prediction with Machine Learning

The data were analyzed as previously described [22,23,39,40]. A data analysis pipeline was developed in ‘R’ (version 3.6.1), which has been used in a number of prior VOC studies [32,33,38]. In short, a 10-fold cross-validation was used for class prediction, with 90% of the data used as a training set and 10% as a test set. The Wilcoxon rank-sum test was used to calculate the *p*-values to identify the 20, 50, and 100 (GC-IMS), and 15 (GC-TOF-MS) most discriminatory features. No chemical identification of the features was carried out for GC-IMS. For GC-TOF-MS, chemical identification of the features was conducted by selecting the chemicals identified in the training set for each fold. The machine learning classification algorithms included in this pipeline were random forest, neural net, sparse logistic regression, Gaussian process, and support vector machine. Machine learning was conducted on the most discriminatory features. For this study, only random forest and sparse logistic regression were considered based on 50 features (GC-IMS) and 15 features (GC-TOF-MS). The predicted probabilities were used to create receiver operator characteristic (ROC) curves for every comparison with their corresponding AUC, sensitivity, specificity, positive predictive value (PPV), and negative predictive value (NPV). A prediction of AUC > 0.75 was considered good.

#### 2.4.6. Metabolite Analysis on GC-TOF-MS Data

Every subgroup comparison resulted in a list of 15 discriminatory metabolites, based on the importance of these metabolites according to the algorithm. Statistical analysis of the metabolite data was performed on the non-zero peak height intensities using IBM SPSS^®^ version 24. The Shapiro–Wilk test was applied to check for normal distribution for each metabolite separately. Since most of the metabolites showed non-normal distribution, all metabolite data were treated as non-normal. Frequency tables were created, and subsequently, the fold change (FC) was calculated based on the median for every metabolite. The FC was normalized by calculating log2(FC). This indicates whether metabolite peak intensities increased or decreased.

## 3. Results

### 3.1. Patient Population

In total, 1013 preterm infants born before 30 weeks of gestation between February 2017 and February 2019 were consecutively assessed for eligibility. Of these, 181 infants (18%) experienced at least one episode of LOS within the first 28 days of life. Based on the inclusion criteria, 121 LOS cases and 121 matched controls were included for GC-IMS analysis, and a subset of 34 cases and 34 controls for GC-TOF-MS analysis. Figure 1 depicts how the inclusion of infants was performed. In total, VOC analysis was performed on 528 fecal samples.

Baseline patient characteristics are listed in Table 1. For infants included in the GC-IMS analysis, there was a significant difference observed between all LOS and CoNS-LOS infants and matched controls in antibiotic exposure from birth to the time of fecal sample analysis. *S. epidermidis* was the most frequently cultured pathogen (47.1%), followed by *E. coli* (14.7%) and *S. aureus* (13.2%). Full microbiology of the blood cultures can be found in Appendix A.

### 3.2. Fecal VOC Patterns

The sparse logistic regression and random forest classification results for GC-IMS and GC-TOF-MS, respectively, are listed in Table 2 and Table 3, respectively. Appendix A show random forest classification for GC-IMS and sparse logistic regression for GC-TOF-MS, respectively. Supplementary Appendix A outline the best ROC curves per comparison for GC-IMS and GC-TOF-MS.

The VOC patterns differed significantly at one and two days before onset of LOS when measured using GC-IMS, as well as when all time points were combined in both GC-IMS and GC-TOF-MS.

For gram-negative LOS, the discriminatory value of GC-IMS was good at three and two days before clinical onset of LOS (AUC [95% CI]: 0.85 [0.74–0.95] and 0.81 [0.70–0.90], respectively; *p* < 0.001), and at one day before onset in the GC-TOF-MS analysis (AUC [95% CI]: 0.78 [0.69–0.87]; *p* < 0.001). When all time points were combined, the discriminatory accuracy was 0.73 and 0.78 for GC-IMS and GC-TOF-MS, respectively (*p* < 0.001). GC-IMS additionally allowed for significant discrimination of *E. coli* LOS from controls at t_−1_ (0.73 [0.53–0.92]; *p* = 0.024) and t_−2_ (0.83 [0.66–1.00]; *p =* 0.002).

The fecal VOC patterns of gram-positive LOS, measured using GC-IMS, but not GC-TOF-MS, could be discriminated from controls at t_−1_ (AUC [95% CI]: 0.78 [0.64–0.90]; *p* = 0.003), and when all time points were combined. For the subcategory of *Staphylococcus aureus* LOS, the fecal VOCs were discriminative from controls at t_−3_ (0.72 [0.50–0.94]; *p* = 0.04). The VOC patterns of coagulase negative LOS were only discriminative when all time points were combined (AUC 0.72; *p* < 0.001 and 0.69; *p* = 0.03 for GC-IMS and GC-TOF-MS, respectively).

### 3.3. Identification of Metabolites using GC-TOF-MS

There were 298 unique volatile metabolites identified in the 158 fecal samples measured with GC-TOF-MS. The classification algorithm selected 15 unique metabolites in every comparison, which together, were the most important in discriminating LOS cases from controls before the onset of clinical symptoms.

Figure 2 depicts the discriminatory metabolites for the subgroups with significantly different VOC profiles when cases were compared with controls. The log2(FC) of the metabolites is provided, indicating whether a metabolite increased or decreased in LOS cases in that comparison. A group of core metabolites was found to be discriminative in almost all significant subgroups. This included 2-methylprop-1-ene, 2-(aziridin-1-yl)ethanamine, propan-2-one, cyclopentane, methoxymethane, propan-2-ol, and dichloromethane. Pathogen-specific discriminative metabolites were identified by focusing on subgroups. Ethyl acetate, ethyl 2-(methylamino)acetate, ethyl 2-hydroxypropanoate, prop-1-ene, butane-2,3-dione, and 2,2,4,4-tetramethylpentane were found to be important for the discrimination of gram-negative LOS versus matched controls. Heptanal was identified as the exclusive metabolite for the discrimination of CoNS-LOS.

## 4. Discussion

In this multicenter case-control study, we identified pathogen-specific fecal VOCs preceding LOS in preterm infants in a longitudinal setting, using two complementary analytical techniques. Observed differences were most evident in gram-negative LOS versus controls, with an accuracy of 0.85 and 0.82 (*p* < 0.001), three days to one day before onset, as measured using GC-IMS and GC-TOF-MS, respectively *E. coli* LOS could be discriminated one and two days before onset (AUC of 0.73 and 0.83, respectively; *p* < 0.05). VOC patterns were also different one day before onset of gram-positive sepsis (AUC of 0.78; *p* < 0.003), and only to a lesser extent for CoNS-LOS. Analysis using GC-TOF-MS revealed a unique set of preclinical pathogen-specific metabolites.

In this study, we aimed to validate previous results illustrating the potential of VOCs as an early, non-invasive biomarker for LOS [32,33,34]. The largest and most recent study performed by our research group (*n* = 127 LOS cases) applied FAIMS technology, which is based on pattern recognition similar to GC-IMS used in the current study, but with a smaller amount of sensors [33]. It was demonstrated that VOC profiles of the last produced stool sample prior to LOS onset can discriminate cases from their matched controls (AUC of 0.77 and 0.74 for gram-negative and gram-positive LOS, respectively) [33]. The increasing performance when focusing on specific LOS pathogens (AUC of 0.86 for *E. coli*)*,* was comparable in the current study. However, the previously observed high accuracy for *S. aureus* sepsis discrimination was not observed in the current data. The recurring low discriminative value of fecal metabolites before CoNS sepsis could be explained by the usual inoculation through indwelling medical devices or dysfunctional skin barriers rather that via the gut [41]. In contrast, gram-negative bacteria colonize the gut, produce toxins, and are capable of effective translocation through the gut lining [42,43]. Gram-negative related LOS, in particular, results in severe clinical symptoms and significantly higher morbidity and mortality. Therefore, the prediction of this subcategory of LOS could have a remarkable impact on neonatal care [44].

As fecal microbiota and VOCs are associated, the distinctive accuracy of VOC measurements in relation to different categories of LOS might reflect the production of unique VOC profiles by different pathogens [45]. Therefore, not only pattern recognition techniques, but also measurements of individual metabolites, were applied in this study. Using GC-TOF-MS, 298 metabolites were identified in the fecal samples. Ethyl acetate––a signature VOC emitted by *E. coli––*was one of the 15 most discriminative metabolites for gram-negative sepsis [45,46,47,48]. Notably, many of the discriminatory metabolites identified in our study have also been described as discriminatory metabolites or as being altered in the feces and breath of patients with (inflammatory) gastrointestinal diseases and colorectal cancer, as shown in Appendix A) [35,49,50,51,52,53,54,55,56,57]. The overlap between discriminative compounds may suggest the presence of shared underlying local and systemic (oxidative or immunological) stress and inflammatory pathways.

Both VOC devices used in the current study have distinct characteristics (pattern recognition versus VOC-identification at a molecular level), and can therefore be considered complementary. Advantages of pattern recognition approaches include the short duration of analysis and relatively low-cost high-throughput capacities, favoring the potential application as a point-of-care tool in daily clinical practice [48]. Additionally, GC-IMS is considered reproducible, and column ageing is less influential than other techniques. Therefore, we decided to analyze all available samples. We first used a pattern recognition technique (GC-IMS) to search for the presence of LOS-specific VOC profiles. Remaining material was additionally analyzed using GC-TOF-MS in order to identify discriminative molecules. The identification of discriminatory molecules could aid in the future development of an e-nose that can predict LOS of different pathogens by recognizing these specific molecules.

This study is strengthened by its multicenter design with prospective data collection and a relatively large number of LOS cases based on a formal power calculation. In addition, the sample preparation and fecal VOC analysis were executed using an evidence-based, standardized, and optimized protocol to release maximal VOC concentrations in the headspace of neonatal samples [58].

This study has some limitations. First, only a limited number of samples was eligible for GC-TOF-MS analysis after GC-IMS was first performed. As a result, the sample size for the GC-TOF-MS analysis, especially for the gram-positive LOS subgroups, was limited. Second, no chemical confirmation of the identified metabolites was performed. Furthermore, although we have tried to limit the risk of including CoNS-contaminated blood cultures by including only cases with CRP concentrations > 10 mg/L, contamination remains a possibility as CRP is a non-specific marker for inflammation. Finally, there was a significant difference between all cases and matched controls regarding antibiotic exposure, which has been suggested to influence fecal VOCs [59]. However, this significant difference was only observed in the GC-IMS CoNS-LOS subgroup and when all cases were combined. Therefore, antibiotic exposure was hypothesized to have only limited influence on the other categories of LOS.

Future studies need to validate the key discriminatory compounds observed in the current study using targeted metabolomics lab-based approaches in larger subgroups. Future research should aim to simultaneously analyze microbiota and (volatile) metabolomics preceding LOS, allowing for computational modelling of metabolic and microbial networks and pathways, as well as predicting the functional activity of the metabolites. This could possibly help elucidate the complex pathogenesis of LOS and support the development of time-sensitive, predictive biomarkers and targeted interventions aimed at prevention of LOS.

To conclude, we confirmed the potential of fecal VOC analysis for the preclinical detection of gram-negative and *E. coli* LOS up to three days prior to clinical onset. Analysis using GC-TOF-MS revealed pathogen-specific VOC patterns and metabolites which, in the future, may contribute to development of a targeted VOC-based point-of-care device to predict LOS preceding onset in a non-invasive manner.

## Figures and Tables

**Figure 1 microorganisms-11-00572-f001:**
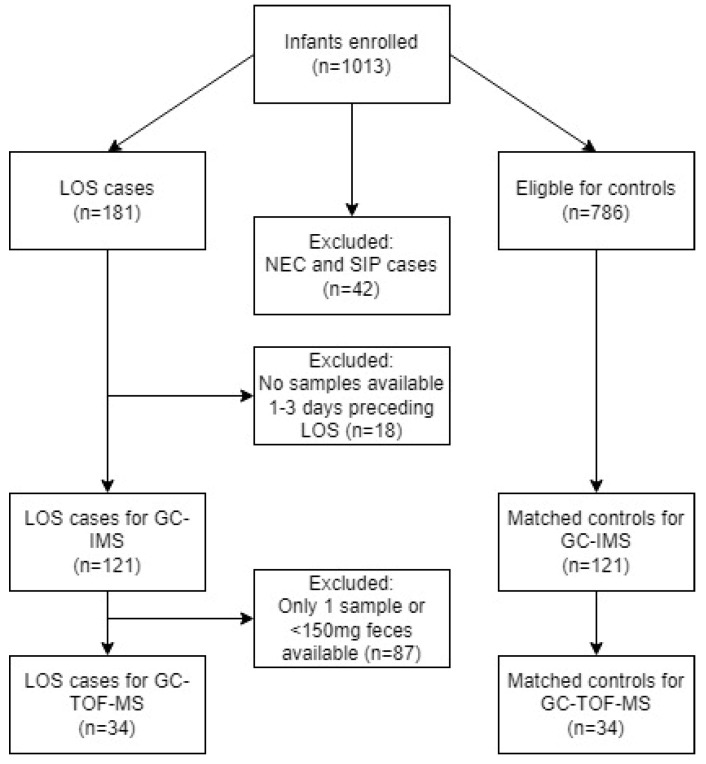
Flow diagram of included preterm infants. A total of 121 LOS cases was included for analysis and matched to 121 controls. Subsequently, 34 LOS cases and controls were eligible for GC-TOF-MS analysis. Abbreviations: GC-IMS, gas chromatography—ion mobility spectrometry; GC-TOF-MS, gas chromatography—time of flight—mass spectrometry; LOS, late-onset sepsis; NEC, necrotizing enterocolitis; SIP, spontaneous intestinal perforation.

**Figure 2 microorganisms-11-00572-f002:**
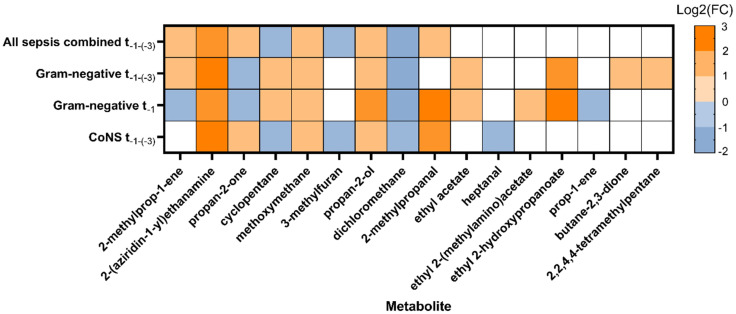
Discriminatory metabolites identified using GC-TOF-MS. The log2(FC) of the median peak height intensities is depicted for every significant comparison, indicating an increase (orange) or decrease (blue) in metabolite abundance in LOS cases. White indicates that the metabolite was not considered discriminatory for that subgroup. Gram-positive LOS is not depicted since there were no significant comparisons. Abbreviations: log2(FC), log2(fold change); LOS, late-onset sepsis.

**Table 1 microorganisms-11-00572-t001:** Demographic characteristics.

	GC-IMS			GC-TOF-MS		
	Case (*n* = 121)	Control (*n* = 121)	*p*-Value ^a^	Case (*n* = 34)	Control (*n* = 34)	*p*-Value ^a^
**Gender**						
Female (*n* (%))	44 (36)	59 (49)	0.05	16 (47)	20 (59)	0.33
**Mode of delivery**						
Vaginal (*n* (%))	67 (55)	70 (58)	0.70	23 (68)	16 (47)	0.09
**GA,** weeks + days (median, (Q1–Q3))	27 + 4 (25 + 6–28 + 6]	27 + 4 (25 + 6–28 + 6)	0.86	26 + 4 (25 + 2–29 + 0)	26 + 4 (25 + 2–29 + 1)	0.69
**BW**, grams (median, (Q1–Q3))	965 (744–1165]	990 (823–1213)	0.42	945 (722–1222)	908 (749–1159)	0.55
**Multiple gestation (*n* (%))**	35 (29)	36 (30)	0.89	11 (32)	17 (50)	0.14
**LOS DoL,** days (median, (Q1–Q3))	9 (7–13]	-	-	14 (8–19)	-	-
**Exposure to AB prior to LOS postnatal age** (*n* (%))	96 (79)	112 (93)	** *0.002* **	30 (88)	31 (91)	0.69
Gram-negative (*n* (%))	25 (86)	31 (97)	0.13	14 (88)	16 (89)	0.90
Gram-positive (*n* (%))	19 (83)	23 (96)	0.15	8 (89)	8 (89)	1.00
CoNS (*n* (%))	50 (81)	56 (95)	** *0.01* **	8 (89)	9 (100)	0.30
**AB duration**, days (median, (Q1–Q3))	3 (2–5)	3 (3–6)	0.20	4 (3–7)	3 (3–5)	0.34
**Feeding type**						
Human milk (*n* (%))	61 (50)	52 (43)		20 (69)	18 (60)	0.72
Formula (*n* (%))	6 (5)	13 (11)	0.15	3 (10)	2 (7)	0.83
Combination (*n* (%))	36 (30)	37 (31)		6 (21)	10 (33)	0.52
**Mortality** (*n* (%))	8 (7)	2 (2)	0.052	3 (11)	1 (4)	0.30

^a^ <0.05 was considered significant. All reported *p*-values were two-sided. Abbreviations: GA, gestational age; BW, birth weight; AB, antibiotics; LOS, late-onset sepsis; CoNS, coagulase-negative *staphylococci*; GC-TOF-MS, gas chromatography—time of flight—mass spectrometry; GC-IMS, gas chromatography—ion mobility spectrometry; N.A., not applicable; Q1, first quartile; Q3, third quartile.

**Table 2 microorganisms-11-00572-t002:** Performance characteristics of gas chromatography—ion mobility spectrometry, as analyzed by sparse logistic regression.

Subgroup	N° of Samples		*p*-Value	AUC (±95% CI)	Sensitivity (±95% CI)	Specificity (±95% CI)	PPV (±95% CI)	NPV (±95% CI)
	Case	Control						
**All categories of late-onset sepsis**								
**3 days before diagnosis (t_−3_)**	82	77	0.151	0.57 (0.49–0.64)	0.59 (0.49–0.68)	0.57 (0.48–0.67)	0.59	0.56
**2 days before diagnosis (t_−2_)**	79	100	0.0001	0.72 (0.66–0.79)	0.59 (0.50–0.69)	0.73 (0.66–0.80)	0.64	0.70
**1 day before diagnosis (t_−1_)**	89	101	0.003	0.63 (0.56–0.69)	0.52 (0.43–0.60)	0.71 (0.63–0.79)	0.61	0.63
**Combined time points (t_−1–(−3)_)**	250	278	0.0001	0.70 (0.66–0.73)	0.58 (0.53–0.63)	0.72 (0.68–0.77)	0.65	0.66
1. Gram-negative late-onset sepsis								
**3 days before diagnosis (t_−3_)**	24	20	0.0001	**0.85 (0.74–0.95) ***	0.88 (0.76–0.97)	0.74 (0.56–0.89)	0.81	0.82
**2 days before diagnosis (t_−2_)**	23	29	0.0001	**0.81 (0.70–0.90) ***	0.65 (0.48–0.82)	0.83 (0.71–0.93)	0.75	0.75
**1 day before diagnosis (t_−1_)**	25	26	0.094	0.64 (0.49–0.76)	0.64 (0.48–0.79)	0.62 (0.45–0.77)	0.62	0.64
**Combined time points (t_−1–(−3)_)**	72	75	0.0001	0.73 (0.66–0.80)	0.64 (0.55–0.73)	0.75 (0.66–0.82)	0.71	0.68
a. *Escherichia coli* late-onset sepsis								
**3 days before diagnosis (t_−3_)**	12	9	0.809	0.61 (0.352–0.87)	0.42 (0.15–0.72)	0.89 (0.52–1.00)	0.80	0.64
**2 days before diagnosis (t_−2_)**	12	14	0.002	**0.83 (0.66–1.00) ***	0.92 (0.62–1.00)	0.64 (0.35–0.87)	0.69	0.90
**1 day before diagnosis (t_−1_)**	14	13	0.024	0.73 (0.53–0.92)	0.64 (0.35–0.87)	0.77 (0.46–0.95)	0.75	0.67
**Combined time points (t_−1–(−3)_)**	38	36	0.379	0.48 (0.34–0.62)	0.53 (0.36–0.69)	0.69 (0.52–0.84)	0.59	0.55
2. Gram-positive late-onset sepsis (excl. coagulase-negative pathogens)								
**3 days before diagnosis (t_−3_)**	17	21	0.472	0.43 (0.28–0.60)	0.35 (0.17–0.56)	0.71 (0.55–0.87)	0.50	0.58
**2 days before diagnosis (t_−2_)**	15	22	0.496	0.43 (0.27–0.60)	0.33 (0.14–0.54)	0.55 (0.36–0.71)	0.33	0.55
**1 day before diagnosis (t_−1_)**	21	20	0.003	**0.78 (0.64–0.90) ***	0.75 (0.58–0.90)	0.80 (0.64–0.94)	0.79	0.76
**Combined time points (t_−1–(−3)_)**	53	63	0.0001	0.70 (0.62–0.78)	0.51 (0.39–0.63)	0.68 (0.59–0.78)	0.57	0.62
a. *Staphylococcus aureus* late-onset sepsis								
**3 days before diagnosis (t_−3_)**	11	12	0.040	0.72 (0.50–0.94)	0.82 (0.48–0.98)	0.58 (0.28–0.85)	0.64	0.78
**2 days before diagnosis (t_−2_)**	8	13	0.598	0.53 (0.25–0.81)	0.50 (0.16–0.84)	0.77 (0.46–0.95)	0.57	0.71
**1 day before diagnosis (t_−1_)**	15	11	0.520	0.50 (0.26–0.73)	0.27 (0.08–0.55)	1.00 (0.72–1.00)	1.00	0.50
**Combined time points (t_−1–(−3)_)**	34	35	0.071	0.60 (0.47–0.74)	0.71 (0.53–0.85)	0.49 (0.31–0.66)	0.57	0.63
3. Coagulase-negative *staphylococci* late-onset sepsis								
**3 days before diagnosis (t_−3_)**	39	36	0.461	0.55 (0.44–0.66)	0.51 (0.38–0.64)	0.61 (0.48–0.74)	0.59	0.54
**2 days before diagnosis (t_−2_)**	41	49	0.476	0.46 (0.35–0.55)	0.41 (0.28–0.55)	0.52 (0.40–0.63)	0.43	0.51
**1 day before diagnosis (t_−1_)**	43	54	0.092	0.60 (0.50–0.70)	0.51 (0.38–0.64)	0.65 (0.54–0.75)	0.54	0.63
**Combined time points (t_−1–(−3)_)**	123	139	0.0001	0.72 (0.66–0.77)	0.67 (0.6–0.74)	0.68 (0.62–0.74)	0.65	0.70

* AUC ≥ 0.75. Abbreviations: AUC, area under curve; NPV, negative predictive value; PPV, positive predictive value; ±95% CI, 95% confidence interval.

**Table 3 microorganisms-11-00572-t003:** Performance characteristics of gas chromatography—time of flight—mass spectrometry, as analyzed using random forest classification.

Subgroup	N° of Samples		*p*-Value	AUC (±95% CI)	Sensitivity (±95% CI)	Specificity (±95% CI)	PPV (±95% CI)	NPV (±95% CI)
	Case	Control						
**All categories of late-onset sepsis**								
**3 days before diagnosis (** **t_−3_** **)**	26	27	0.810	0.48 (0.34–0.62)	0.46 (0.30–0.63)	0.59 (0.43–0.75)	0.52	0.53
**2 days before diagnosis (** **t_−2_** **)**	26	25	0.231	0.60 (0.45–0.73)	0.69 (0.54–0.83)	0.48 (0.31–0.65)	0.58	0.60
**1 day before diagnosis (** **t_−1_** **)**	27	27	0.729	0.53 (0.40–0.67)	0.59 (0.44–0.75)	0.63 (0.48–0.79)	0.62	0.61
**Combined time points (t_−1–(−3)_)**	79	79	0.001	**0.77 (0.71–0.83) ***	0.68 (0.60–0.77)	0.73 (0.65–0.82)	0.72	0.70
1. Gram-negative late-onset sepsis								
**3 days before diagnosis (** **t_−3_** **)**	11	13	0.060	0.73 (0.54–0.90)	0.55 (0.29–0.78)	0.85 (0.67–1.00)	0.75	0.69
**2 days before diagnosis (** **t_−2_** **)**	16	10	0.370	0.61 (0.39–0.82)	0.88 (0.73–1.00)	0.30 (0.08–0.56)	0.67	0.60
**1 day before diagnosis (** **t_−1_** **)**	11	12	0.010	**0.82 (0.64–0.95) ***	0.64 (0.38–0.88)	0.92 (0.77–1.00)	0.88	0.73
**Combined time points (t_−1–(−3)_)**	38	35	0.0001	**0.78 (0.69–0.87) ***	0.74 (0.62–0.85)	0.71 (0.58–0.83)	0.74	0.71
2. Gram-positive late-onset sepsis (excl. coagulase-negative *staphylococci* pathogens)								
**3 days before diagnosis (** **t_−3_** **)**	6	9	0.724	0.44 (0.17–0.70)	0.17 (0.00–0.43)	0.56 (0.25–0.82)	0.20	0.50
**2 days before diagnosis (** **t_−2_** **)**	4	7	0.449	0.64 (0.25–1.00)	0.75 (0.33–1.00)	0.86 (0.60–1.00)	0.75	0.86
**1 day before diagnosis (** **t_−1_** **)**	10	7	0.807	0.54 (0.29–0.80)	0.80 (0.56–1.00)	0.14 (0.00–0.40)	0.57	0.33
**Combined time points (t_−1–(−3)_)**	20	23	0.846	0.52 (0.37–0.67)	0.40 (0.23–0.59)	0.65 (0.48–0.81)	0.50	0.56
3. Coagulase-negative *staphylococci* late-onset sepsis								
**3 days before diagnosis (** **t_−3_** **)**	9	5	0.841	0.47 (0.06–0.83)	0.89 (0.70–1.00)	0.40 (0.00–0.80)	0.73	0.67
**2 days before diagnosis (** **t_−2_** **)**	9	7	0.223	0.32 (0.08–0.60)	0.43 (0.13–0.75)	0.67 (0.38–0.91)	0.50	0.60
**1 day before diagnosis (** **t_−1_** **)**	7	8	0.643	0.52 (0.23–0.79)	0.43 (0.13–0.75)	0.63 (0.33–0.90)	0.50	0.56
**Combined time points (t_−1–(−3)_)**	23	22	0.026	0.69 (0.57–0.83)	0.74 (0.59–0.88)	0.68 (0.52–0.85)	0.71	0.71

* AUC ≥ 0.75. Abbreviations: AUC, area under curve; NPV, negative predictive value; PPV, positive predictive value; ±95% CI, 95% confidence interval.

## Data Availability

The data is publicly available at DOI: 10.6084/m9.figshare.19323905.

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
