# Peer review of "Fecal Volatile Metabolomics Predict Gram-Negative Late-Onset Sepsis in Preterm Infants: A Nationwide Case-Control Study"

_microorganisms, 2023, doi:10.3390/microorganisms11030572_

Round 1

Reviewer 1 Report

This is a very important study with interesting findings. The usage of VOCs will open up a big area in research for early disease detection.  However, there are some vital points needed to be added or revised before considering it further. Here are the comments needed to be addressed:

- The introduction requires further information for different areas. 

1. The cellular metabolic processes and their changes are manifested in emitted volatile organic compound (VOC) compositions of different diseased cells, such as cancer cells will open up the possibilities of early detection techniques. The INTRODUCTION section needs information and background on metabolic changes and VOCs and their role in the detection of diseases (ie. cancer). Please also highlight in the section, the other techniques besides GC-iMS, such as LC-MS and cyborg techniques. Some recent citations needed to be added, such as the following one from a very nice journal, explaining a technique of insect brain-based cancer VOC detection using AI techniques. 

https://doi.org/10.1016/j.bios.2022.114814 

- Please separate the aims and objectives of the study from the second paragraph in the "Introduction" section.

Reviewer 2 Report

1. How defined the quantity of human milk?

2. Maybe the distinction must be between exclusive HM and FM and Combined . The microbiome established in idifferent way for the exclusive breastfeeding babies (doi:10.3390/children9020154 )

3. In the samples of the study, were the populations of the intestinal flora also studied?

4. there is a correlation of the metabolites detected in the feces with the prognosis of the infection ?
